# Major bleeding risk and mortality associated with antiplatelet drugs in real-world clinical practice. A prospective cohort study

Jacques Bouget[1], Frédéric Balusson[1], Damien Viglino[2], Pierre-Marie Roy[3,4], Karine Lacut[5], Laure Pavageau[6], Emmanuel Oger[1]*

**1** EA 7449 [Pharmacoepidemiology and Health Services Research] REPERES, Univ Rennes, CHU Rennes, Rennes, France, **2** Emergency Department, Grenoble-Alpes University Hospital, Grenoble, France, **3** Emergency Department, Centre Hospitalier Universitaire, Institut MITOVASC, Université d'Angers, Angers, France, **4** F-CRIN INNOVTE, France, **5** CIC 1412, Université de Bretagne Loire, Université de Brest, INSERM CIC 1412, CHRU de Brest, Brest, France, **6** Emergency Department, University Hospital, Nantes, France

* emmanuel.oger@univ-rennes1.fr

**Data Availability Statement:** Under French law and regulations, patient-level data from the SNIIRAM database cannot be made publicly available. Data access requests may be directed to

## Abstract

### Background

Major bleedings other than gastrointestinal (GI) and intracranial (ICH) and mortality rates associated with antiplatelet drugs in real-world clinical practice are unknown. The objective was to estimate major bleeding risk and mortality among new users of antiplatelet drugs in real-world clinical practice.

### Methods and findings

A population-based prospective cohort using the French national health data system (SNIIRAM), identified 69,911 adults living within five well-defined geographical areas, who were new users of antiplatelet drugs in 2013–2015 and who had not received any antithrombotics in 2012. Among them, 63,600 started a monotherapy and 6,311 a dual regimen. Clinical data for all adults referred for bleeding was collected from all emergency departments within these areas, and medically validated. Databases were linked using common key variables. The main outcome measure was time to major bleeding (GI, ICH and other bleedings). Secondary outcomes were death, and event-free survival (EFS). Hazard ratios (HR) were derived from adjusted Cox proportional hazard models. We used Inverse Propensity of Treatment Weighting as a stratified sensitivity analysis according to the antiplatelet monotherapy indication: primary prevention without cardiovascular (CV) risk factors, with CV risk factors, and secondary prevention. We observed 250 (0.36%) major haemorrhages, 81 ICH, 106 GI and 63 other types of bleeding. Incidences were twice as high in dual therapy as in monotherapy. Compared to low-dose aspirin ($\leq$ 100 mg daily), high-dose (> 100 up to 325 mg daily) was associated with an increased risk of ICH (HR = 1.80, 95%CI 1.10 to 2.95). EFS was improved by high-dose compared to low-dose aspirin (1.41, 1.04 to 1.90 and 1.32, 1.03 to 1.68) and clopidogrel (1.30, 0.73 to 2.3 and 1.7, 1.24 to 2.34) respectively in primary prevention with and without CV risk factors.

info@indsante.fr. A list of relevant data set names to request the minimal data set are provided in the Supporting Information file. The statistical code is available from Figshare (DOI: 10.6084/m9.figshare.12581999.v1).

**Funding:** This study was supported by the National Clinical Research Hospital Program of the French Ministry of Health (PHRC-12-009-0243). The funder had no role in the study design, data collection, analysis, or interpretation, in the writing of the article, or in the decision to submit it for publication.

**Competing interests:** The authors have declared that no competing interests exist.

## Conclusion

The incidence of major bleeding and mortality was low. In monotherapy, low-dose aspirin was the safest therapeutic option whatever the indication.

## Trial registration

NCT02886533.

## Introduction

Antiplatelet drugs are used for primary and secondary prevention of ischemic cardiovascular events [1]. Long-term low-dose aspirin (<325 mg daily) or clopidogrel are recommended for patients with symptomatic peripheral artery disease, before and after peripheral arterial bypass surgery or percutaneous transluminal angioplasty [2], and for patients with non-cardio-embolic ischemic stroke and transient ischemic attack [3,4]. For patients with acute coronary syndromes who undergo a percutaneous coronary intervention with stent placement, dual antiplatelet therapy for the first year is recommended, with low-dose aspirin in combination with clopidogrel, ticagrelor or prasugrel. Among patients with coronary stents, low-dose aspirin plus clopidogrel is recommended for 12 months [5,6]. Aspirin is also used for primary prevention among patients with an increased risk of cardiovascular events or among patients with diabetes mellitus, although these indications appear more controversial [7,8].

The longer life expectancy of the population, and consequently the increase in the prevalence of these ischemic cardiovascular diseases, have led to an increase in the proportion of patients who require treatment with antiplatelet drugs, especially among the elderly [9].

Whatever their indications and the regimen used, antiplatelet drugs are associated with a risk of bleeding, leading to substantial morbidity and mortality. In monotherapy, the increased risk of gastrointestinal bleeding has been commonly reported [10], in contrast to the risk of intracranial bleeding which appears more controversial [11]. In dual therapy, the risk of major non-fatal bleeding is increased whatever the dual antiplatelet regimen [12]. Major bleedings other than GI and intracranial have almost never been reported.

Mortality rates associated with antiplatelet drugs seem dependent on the indications: in two meta-analyses of randomized clinical trials [7,13], and in contemporary US cohorts [14], a similar all-cause mortality was reported with aspirin alone versus control in primary prevention. A decrease in cardiovascular mortality with aspirin versus control was observed in secondary prevention [13]. Various mortality rates have been reported when several antiplatelet drugs were compared to each other in randomized trials [15–17]. To our knowledge, mortality rates associated with antiplatelet treatment in real-world clinical practice are not known.

Our aim was to estimate the incidence of major bleeding and mortality associated with antiplatelet drugs, irrespective of the indication, in real-world clinical practice. Only new users of antiplatelet drugs were considered in this prospective study.

## Methods

### Study design and participants

The study design has previously been reported [18]. Briefly, a prospective population-based cohort study was set up linking the French Health Insurance Database (SNIIRAM) to data from all emergency departments located within five well-defined areas around five large

French cities (Angers, Brest, Grenoble, Nantes and Rennes). The study received regulatory approval (*Commission Nationale de l'Informatique et des Libertés* CNIL, DR-2013-488 with subsequent substantial changes DR-2016-489). The need for informed consent was waived.

The SNIIRAM anonymously and comprehensively links the universal healthcare reimbursement database (DCIR) to the French hospital discharge database (PMSI). The DCIR contains anonymous individual data on all reimbursements for health expenditure, including drugs. This database does not provide any direct information on the medical indication for each reimbursement. The PMSI provides hospital discharge diagnoses (ICD-10 code) as well as details on medical acts. Thus, all adult subjects (> 18 years) living in the five above-mentioned areas, affiliated to the French universal Health Insurance System, and with at least one reimbursement for an antiplatelet drug (aspirin, clopidogrel, prasugrel, ticagrelor) in 2013–2015 were identified.

From emergency departments, clinical data for all adult subjects living in the five above-mentioned areas and referred between January 1, 2013 and December 31, 2015 for bleeding while receiving an antiplatelet drug was prospectively collected, focusing on the type of bleeding. Their demographics were collected (month and year of birth, gender) as well as their date of hospital admission. In each emergency department, the medical referent validated the final inclusion of all records of severe bleeding detected.

The SNIIRAM and clinical databases were linked using common key variables (date of birth (month, year), gender, date of hospital entry and discharge, type of antiplatelet drug, and care facility involved).

Only "new users" defined as having had neither an antiplatelet drug delivery nor any other antithrombotic drug delivery in 2012 were analysed.

Patient characteristics were collected first. We calculated the adapted Charlson's index (see S1 Table in S1 File) from co-morbidities and co-medication retrieved from the SNIIRAM database [19].

## Outcomes

The primary outcome was major bleeding. Secondary outcomes were death and major-bleeding-event-free survival.

Major bleeding was defined from at least one of the following criteria: unstable haemodynamics (systolic arterial pressure < 90 mmHg or mean arterial pressure < 65 mm Hg) or haemorrhagic shock, uncontrollable bleeding, need for transfusion or haemostatic procedure (embolization, endoscopic procedure, surgery). The location or the symptoms then defined the type: intracranial haemorrhage (ICH), acute gastrointestinal (GI) bleeding, and other major bleeding in life-threatening locations—intraspinal, intraocular, retroperitoneal, pericardial, thoracic, intra-articular, or intramuscular haematoma with compartment syndrome. We also considered the following as "other" major bleeding events: epistaxis with at least two procedures of nasal packing, and hematuria when bleeding lasted more than 12 hours despite bladder washing. There was a slight alteration with respect to the International Society on Thrombosis and Haemostasis (ISTH) classification of major bleeding events, because in our dataset no information was available on haemoglobin levels [20]. Major bleeding events were further classified into three classes: intracranial haemorrhage (ICH), GI bleeding and other bleeding events. For each type of bleeding event, censoring was based on death or any other type of major bleeding event or end of study follow-up, whichever came first.

Major-bleeding-event-free survival was defined as time to major bleeding or death or end of study follow-up, whichever came first.

## Main exposure to antiplatelet drugs

Antiplatelet drug exposure was defined as the first observed delivery, and patients were categorised as new users of either low-dose aspirin (acetylsalicylic acid, ASA ≤ 100 mg daily), high-dose ASA (more than 100 up to 325 mg daily), or clopidogrel for mono-therapies, and ASA + clopidogrel or prasugrel or ticagrelor for dual therapies. Patients were censored if they discontinued antiplatelet therapy (using the date of end of supply) or if they switched to another antithrombotic drug (oral anticoagulant, parenteral anticoagulant). The date of first delivery of the new treatment following a switch was considered as the end of the supply of the antiplatelet drug. The same applied in case dual prescription. Patients were likewise censored in case of death or major bleeding, or if they had moved away from the area, or had reached the end of study follow-up, whichever occurred first.

The indication for antiplatelet prescription was derived from the main discharge diagnosis and/or medical acts performed in the 30 days before the first observed issue of antiplatelet therapy (see S2 Table in S1 File). Three groups of patients according to antiplatelet monotherapy indications were defined: 1) patients with antiplatelet drug for secondary prevention (coronary disease, carotid artery disease, peripheral artery disease, stroke, valvular heart disease, miscellaneous diseases), 2) patients with antiplatelet drugs for primary prevention with identified cardiovascular (CV) risk factors (arterial hypertension, diabetes, renal disorders, lipid-lowering drugs), and 3) patients with antiplatelet drugs for primary prevention without any identified CV risk factors (no factors identified by the SNIIRAM database).

## Statistical analysis

Crude incidence rates (IR) were calculated for the first bleeding episode per 10,000 person-months according to the type of bleeding, the type of antiplatelet regimen, and the duration of use (under 6 months, 6 to 12 months or over one year for monotherapy, and under 3 months, 3 to 6 months and over 6 months for dual therapy).

Cox proportional hazard regression analyses were conducted for each type of bleeding event to determine hazard ratios for different antiplatelet regimens compared to low-dose ASA (≤100 mg) used as a reference for monotherapy, and compared to ASA + clopidogrel as a reference for dual therapy, adjusted on known patient characteristics: age, gender, diabetes, arterial hypertension, previous bleeding, medication use (lipid-lowering drug, proton pump inhibitor, NSAID) and co-morbidities using the adapted Charlson index.

For secondary outcomes, Cox proportional hazard regression models were run to estimate death and event-free survival for any antiplatelet drug in mono or dual therapy, irrespective of bleeding type.

For antiplatelet monotherapy, a sensitivity analysis stratified by indication was conducted. We first used the stabilized inverse probability of treatment weighting (SIPTW) based on the propensity score [21]. Separate analyses were performed for the three main indications of antiplatelet monotherapy: primary prevention without CV factors, primary prevention with CV factors and secondary prevention. The weights were truncated by resetting the value of weights greater (or lower) than the 99th (1st) percentile to the value of 99th (1st) percentile [22]. Weighted Kaplan-Meier survival functions were generated for death and event-free survival for these 3 indications.

All statistical tests were two-tailed and P-values <0.05 were considered significant. Statistical analyses were performed using SAS software 9.4 (SAS Institute, Cary, N.C., USA).

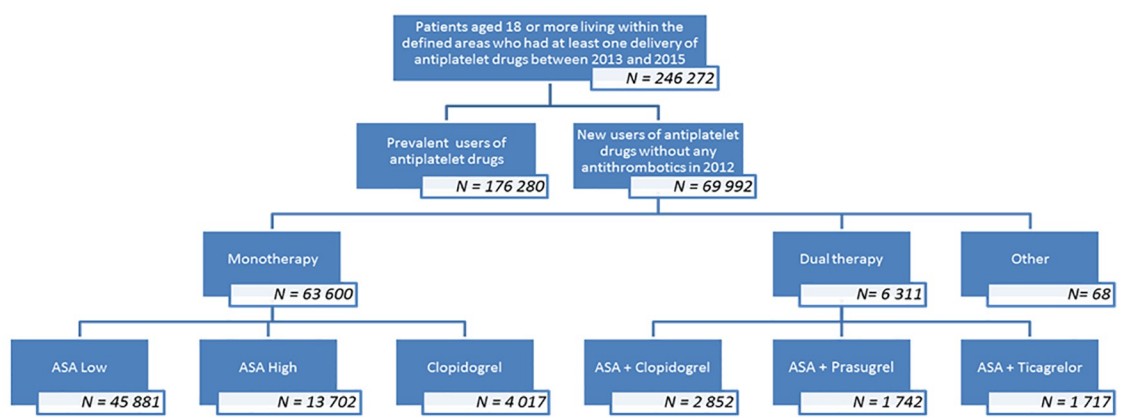

**Fig 1. Inclusion flow chart.**

## Results

### Cohort characteristics

Between 2013 and 2015, 246,272 patients aged 18 or over living within the defined areas had at least one delivery of antiplatelet drugs. Only 69,992 patients were new users of antiplatelet drugs and constituted the study cohort (Fig 1, flow chart). Antiplatelet monotherapy accounted for 91% of the cohort. Sixty-eight patients receiving combinations of antiplatelet drugs other than mono or dual therapy were excluded from the final analysis.

Characteristics of the population according to antiplatelet therapy (mono and dual therapy) are reported in Table 1. Among patients under different antiplatelet monotherapy regimens, the median age ranged from 65 to 67 years and 19% to 21% of the patients were over 80. Out of 5,674 patients classified as having coronary artery disease as presumed indication for antiplatelet monotherapy, 1,026 had been hospitalized (out of them 705 suffered from ACS) and 4,048 had ambulatory care for angina pectoris identified through stress tests. Then 705 out of 5,674 patients (12.4%) suffered from ACS. Whatever the antiplatelet regimen, 85% to 95% of patients had an adapted Charlson comorbidity index of 2 or under. Patients with dual therapy including prasugrel or ticagrelor were younger, and respectively 96% or 97% had a coronary disease with current use of lipid-lowering drugs. Twenty-seven to 32% of the patients with different antiplatelet monotherapy regimens were using concomitant antiulcer medication, and 56% to 59% of the patients with different dual therapy regimens.

The distribution of CV risk factors according to antiplatelet therapy among patients treated for primary prevention is shown in S3 Table in S1 File. Among patients with low-dose ASA, high-dose ASA and clopidogrel monotherapy, 53%, 63% and 59% respectively had one risk factor, 36%, 29% and 32% respectively had 2 risk factors and 11%, 7% and 9% had at least 3 risk factors.

During the study period, a total of 250 (0.36%) patients experienced a first major bleeding event: 81 patients (0.12%) with ICH, 106 (0.15%) with GI bleeding, and 63 (0.09%) with other major bleeding.

Bleeding sites according to antiplatelet drugs are reported in the supplementary material in S4 Table in S1 File Major bleeding was about twice as frequent among patients with dual therapy compared to monotherapy, in particular with the ASA + prasugrel or ticagrelor combinations. GI was the main site of bleeding, whatever the antiplatelet regimen, especially the upper

**Table 1. Characteristics of subjects by drug exposure.**

| Characteristics | Monotherapy | | | Dual therapy | | |
|---|---|---|---|---|---|---|
| | ASA Low (≤ 100 mg) | ASA High (> 100–325 mg) | Clopidogrel | ASA + Clopidogrel | ASA + Prasugrel | ASA + Ticagrelor |
| | N = 45 881 | N = 13 702 | N = 4 017 | N = 2 852 | N = 1 742 | N = 1 717 |
| Age, median (Q1-Q3) | 65 (53–76) | 66 (54–77) | 67 (58–78) | 68 (57–80) | 57 (49–64) | 61 (52–72) |
| > 80 years | 8 723 (19.0) | 2 834 (20.7) | 844 (21.0) | 735 (25.8) | 8 (0.5) | 196 (11.4) |
| Gender, female | 25 416 (55.4) | 6 917 (50.5) | 1 792 (44.6) | 1 087 (38.1) | 266 (15.3) | 420 (24.5) |
| Context | | | | | | |
| Primary prevention without risk factors | 15 561 (33.9) | 3 499 (25.5) | 701 (17.5) | 148 (5.2) | 5 (0.3) | 12 (0.7) |
| Primary prevention with risk factors £ | 18 007 (39.3) | 3 393 (24.8) | 1 290 (32.1) | 390 (13.7) | 43 (2.5) | 56 (3.3) |
| Secondary prevention | 12 313 (26.8) | 6 810 (49.7) | 2 026 (50.4) | 2 314 (81.1) | 1 694 (97.2) | 1 649 (96.0) |
| Coronary diseases | 4 841 (10.6) | 545 (4.0) | 288 (7.2) | 2 029 (71.1) | 1 694 (97.2) | 1 647 (95.9) |
| Carotid | 2 783 (6.1) | 1 059 (7.7) | 282 (7.0) | 20 (0.7) | - | 1 (0.1) |
| PVD | 3 592 (7.8) | 743 (5.4) | 1 183 (29.4) | 136 (4.8) | - | 1 (0.1) |
| Stroke | 863 (1.9) | 4 334 (31.6) | 266 (6.6) | 94 (3.3) | - | - |
| Valvular heart disease | 203 (0.4) | 121 (0.9) | 6 (0.1) | 35 (1.2) | - | - |
| Other ¥ | 31 (0.1) | 8 (0.1) | 1 (0) | - | - | - |
| Comorbidities † | | | | | | |
| Diabetes mellitus | 9 705 (21.2) | 1 457 (10.6) | 727 (18.1) | 408 (14.3) | 204 (11.7) | 213 (12.4) |
| Hematologic or immune disease | 894 (1.9) | 372 (2.7) | 84 (2.1) | 104 (3.6) | 24 (1.4) | 20 (1.2) |
| Previous bleeding | 546 (1.2) | 214 (1.6) | 60 (1.5) | 53 (1.9) | 8 (0.5) | 9 (0.5) |
| Medication use (Current 3 months after) | | | | | | |
| Lipid-lowering drug | 18 047 (39.4) | 6 363 (46.4) | 2 060 (51.3) | 2 247 (78.8) | 1 662 (95.4) | 1 618 (94.2) |
| Antiulcer agent | 12 728 (27.7) | 4 211 (30.7) | 1 318 (32.8) | 1 614 (56.6) | 992 (56.9) | 1 013 (59.0) |
| NSAID | 3 992 (8.7) | 1 002 (7.3) | 301 (7.5) | 88 (3.1) | 52 (3.0) | 38 (2.2) |
| Modified Charlson comorbidity index § | | | | | | |
| 0 | 35 470 (77.3) | 6 664 (48.6) | 2 436 (60.6) | 1 704 (59.7) | 1 436 (82.4) | 1 337 (77.9) |
| 1–2 | 8 439 (18.4) | 5 075 (37.0) | 1 305 (32.5) | 868 (30.4) | 277 (15.9) | 318 (18.5) |
| 3–4 | 1 323 (2.9) | 1 618 (11.8) | 214 (5.3) | 206 (7.2) | 22 (1.3) | 46 (2.7) |
| ≥ 5 | 649 (1.4) | 345 (2.5) | 62 (1.5) | 74 (2.6) | 7 (0.4) | 16 (0.9) |

Values are numbers (percentages) unless stated otherwise; ASA Low stands for low-dose aspirin (≤ 100 mg daily), and ASA High for > 100 up to 325 mg daily;

* based on hospital discharge main diagnosis (according to ICD-10 or medical act classification in the month before the index date);

† based on hospital discharge diagnosis (according to ICD-10 or co-medications (ATC system) in the previous year, see supplementary appendix, S1 Table in S1 File for details; see supplementary appendix, S2 Table in S1 File for details;

§ as defined by Bannay et al, see supplementary appendix;

£ = Arterial hypertension, Lipid-lowering drug, Diabetes, Renal disorders;

¥ = Eclampsia, Hematological disorders, retinopathy. ASA stands for acetylsalicylic acid, PVD for peripheral vascular disease. NSAID for non-steroid anti-inflammatory drug

GI tract. Among the 81 ICH observed, 43 were intracerebral haemorrhages and 31 extra- or subdural hematomas. Hematuria and oropharyngeal bleeding were the most frequent other major types of bleeding.

## Incidence rates for major bleeding events according to exposure to antiplatelet monotherapies

Crude incidence rates of major bleeding according to antiplatelet monotherapy and time period are reported in Fig 2 (Panel A). Values are detailed in the supplementary material in

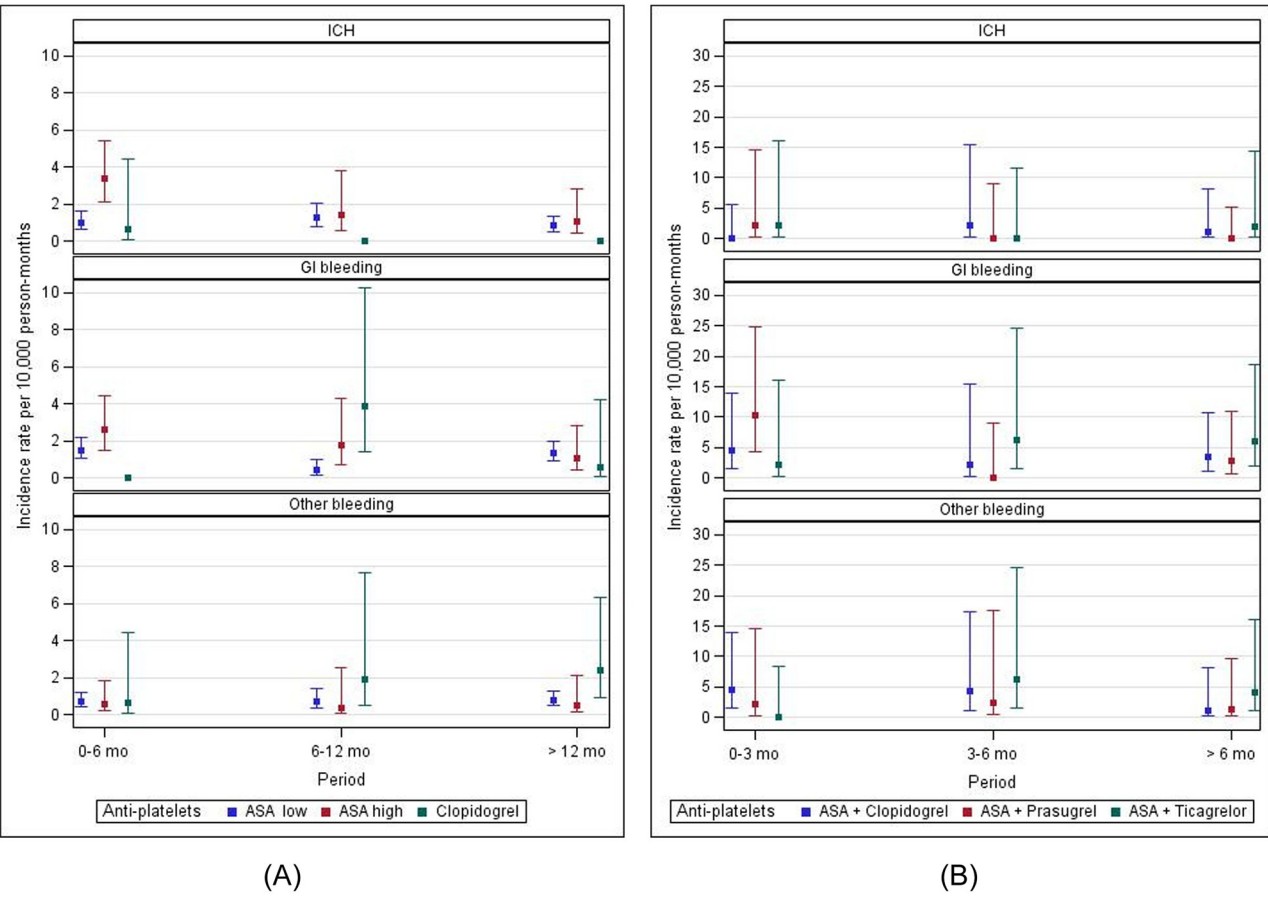

**Fig 2. Crude incidence rates for major bleeding according to antiplatelet monotherapy and time period.** Panel A: Monotherapy. Panel B: Dual antiplatelet regimen.

S5A Table in S1 File. Altogether, incidence rates were low. The incidence rates of intracranial haemorrhage (ICH) and gastrointestinal (GI) bleeding were higher in the first six months of treatment with high-dose ASA, and decreased regularly with time. This observation was not found with low-dose ASA nor with clopidogrel.

## Associations between exposure to antiplatelet monotherapy and outcomes

Forest plots showing adjusted hazard ratios (HRs) for each first bleeding episode, death and event-free survival for antiplatelet monotherapy, compared to low-dose ASA as a reference are presented in Fig 3.

Compared to low-dose ASA, high-dose ASA was associated with a significantly increased risk of ICH (HR = 1.80, 95%CI 1.10 to 2.95) and a non-significant increased risk of GI bleeding (HR = 1.54, 95%CI 0.93 to 2.55). Clopidogrel was associated with a non-significant increased risk of other major bleeding events (HR = 2.03, 95%CI 0.90 to 4.57). No significant differences were reported concerning death or EFS (Fig 3).

Stratified sensitivity analyses were conducted across three indications: primary prevention without CV risk factors (n = 19,761), primary prevention with CV risk factors (n = 22,690), and secondary prevention (n = 21,149). Baseline characteristics are reported in supplementary S6 Table in S1 File.

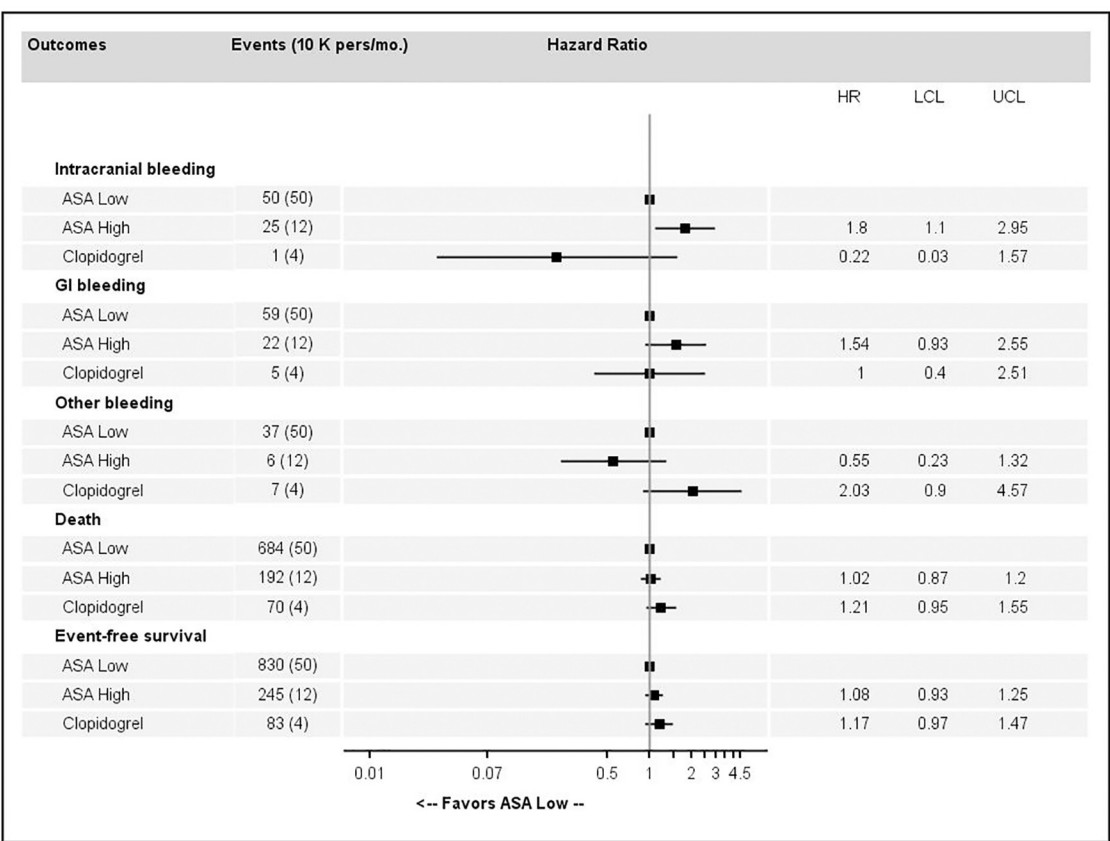

**Fig 3. Association estimates between major bleeding, death or event-free survival and high-dose acetylsalicylic acid (ASA) or clopidogrel compared to low-dose ASA.**

Association estimates between major bleeding, death or event-free survival and high-dose ASA or clopidogrel compared to low-dose ASA as a reference are reported as follows: in Fig 4 for primary prevention without CV risk factors, in Fig 5 for primary prevention with CV risk factors and in Fig 6 for secondary prevention. EFS was significantly improved by high-dose ASA (HR = 1.41, 95%CI 1.04 to 1.90 and HR = 1.32, 95%CI 1.03 to 1.68) and clopidogrel (HR = 1.30, 95%CI 0.73 to 2.3 and HR = 1.7, 95%CI 1.24 to 2.34) in primary prevention among patients without CV risk factor and in primary prevention among patients with CV risk factor respectively, compared to low-dose ASA. In secondary prevention, clopidogrel was significantly better than low-dose ASA in terms of EFS (HR = 0.58, 95%CI 0.38 to 0.88) but there were a few bleeding events (Fig 6). In primary prevention without CV risk factors (Fig 4), high-dose ASA was associated with a higher risk of ICH and GI bleeding. In primary prevention with CV risk factors (Fig 5), high-dose ASA and clopidogrel were associated with a higher risk of death and lower EFS. In secondary prevention (Fig 6), clopidogrel was associated with a lower risk of death and better EFS.

Weighted Kaplan-Meyer survival functions for EFS according to antiplatelet therapy and indications are shown in Fig 7. In primary prevention without CV risk factor, low-dose ASA had a significantly better survival function than high-dose ASA and clopidogrel (p = 0.051) (Fig 7 panel A). Clopidogrel had a significantly better EFS in secondary prevention (p = 0.042) (Fig 7 panel C) and a poorer survival function in primary prevention with CV risk factors

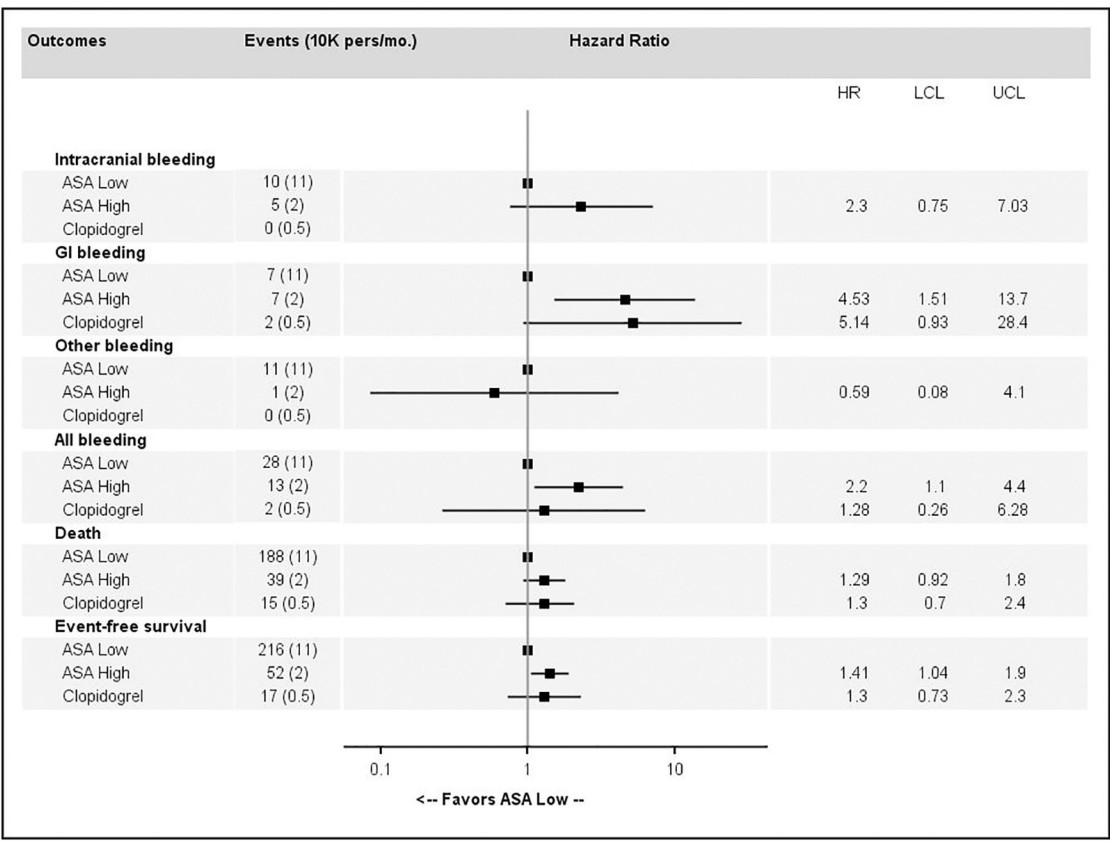

**Fig 4. Association estimates between major bleeding, death or event-free survival and high-dose acetylsalicylic acid (ASA) or clopidogrel compared to low-dose ASA for primary prevention without cardiovascular risk factors.**

(p<0.001) (Fig 7 panel B). Weighted Kaplan-Meyer survival functions for death according to antiplatelet monotherapy and indications are reported in S1 Fig. There was significantly poorer survival with clopidogrel compared to low-dose ASA in primary prevention with CV risk factors (p<0.001).

## Incidence rates for major bleeding according to exposure to dual antiplatelet regimens

Crude incidence rates for major bleeding according to dual antiplatelet regimens and time period are reported in Fig 2 (Panel B). Values are detailed in supplementary S5B Table in S1 File. Incidence rates for all major bleeding events were higher in dual regimens than in mono-therapy. There were no significant variations in the incidence rates of any major bleeding event with the different dual regimens, whatever the duration of the treatment.

## Association between exposure to dual antiplatelet regimen and outcomes

Forest plots showing adjusted hazard ratios (HR) for each first bleeding episode, death and EFS for dual antiplatelet regimens compared to ASA + clopidogrel as a reference are reported in Fig 8. No significant differences in the different types of major bleeding were reported between the combinations of ASA + prasugrel or ticagrelor.

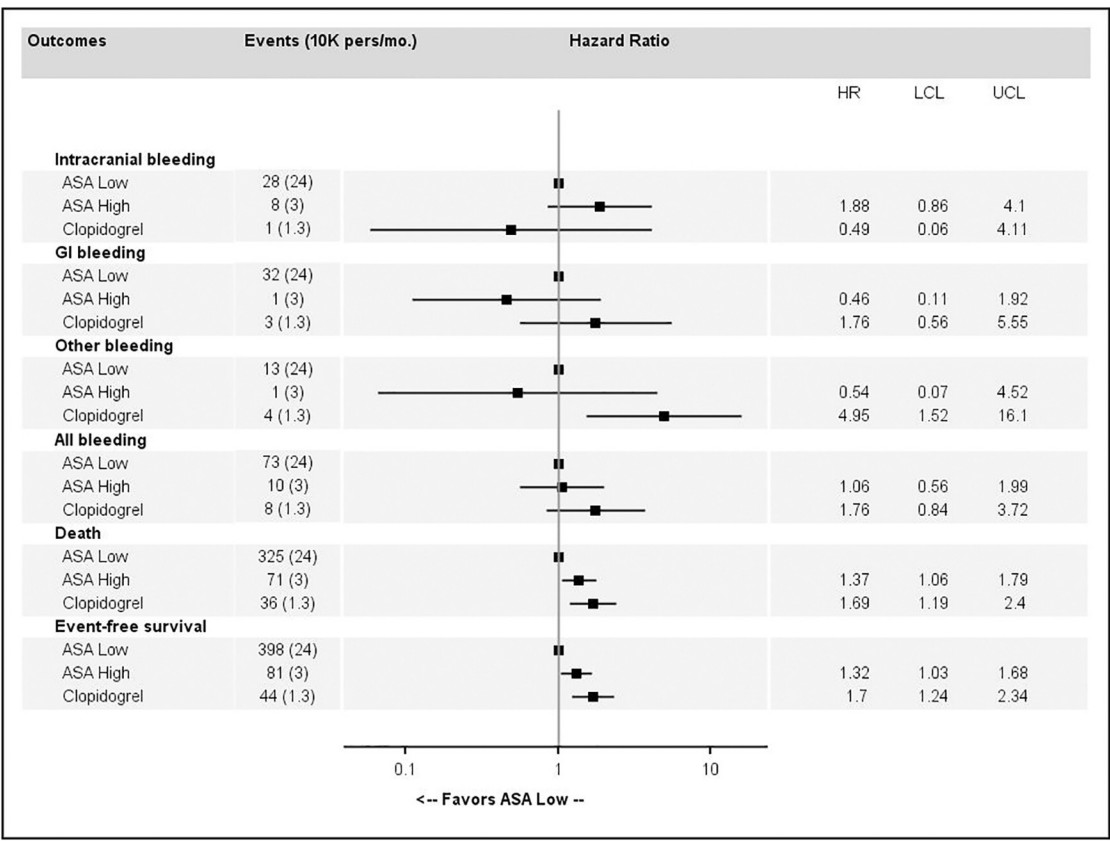

**Fig 5. Association estimates between major bleeding, death and or event-free survival and high-dose acetylsalicylic acid (ASA) or clopidogrel compared to low-dose ASA for primary prevention with cardiovascular risk factors.**

## Discussion

### Principal findings

Using an administrative healthcare database linked to prospectively collected clinical data from emergency departments, our population-based cohort study showed low incidence rates of major bleeding and low mortality rates among new antiplatelet drug users. A 2- to 3-fold increased risk of gastrointestinal bleeding and other major bleeding was observed in dual regimens compared to monotherapy, with a better safety profile for low-dose ASA ($\leq$ 100 mg) among monotherapies. There were no significant differences across dual regimens. We observed very few events in patients prescribed with dual regimens, keeping in mind only major bleeding were accounted for; given these low incidences we had a very low power to detect any difference.

### Strengths and weaknesses

This large, comprehensive study is original for its ability to directly compare different antiplatelet drugs, including the different doses and regimens, encompassing all indications, to explore important, common safety outcomes. Patients exposed to anticoagulants combined with antiplatelet therapy were not included in this analysis. Analysing major bleeding minimizes the bias that affects the external validity of studies focusing on hospital data only, where

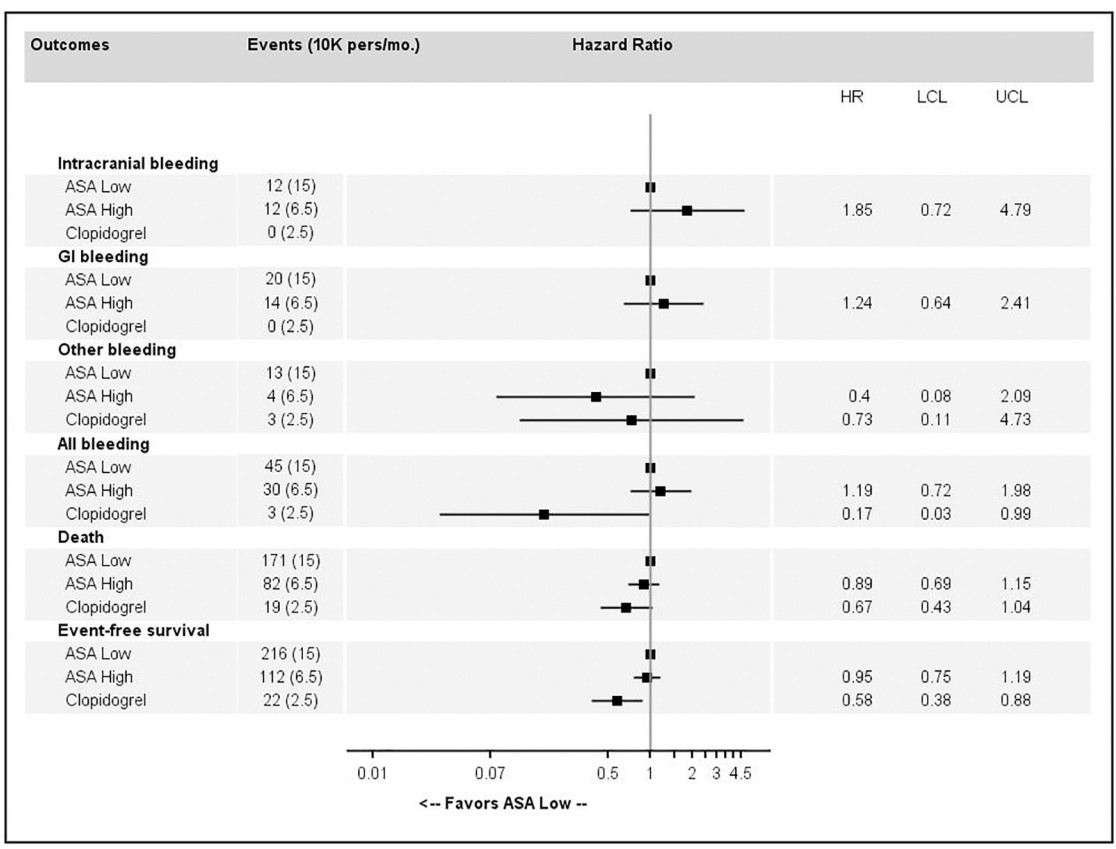

**Fig 6. Association estimates between major bleeding, death and or event-free survival and high-dose acetylsalicylic acid (ASA) or clopidogrel compared to low-dose ASA for secondary prevention.**

patients diagnosed and managed in primary care are not included. An important point is that this study retrieved and linked data from a prospective multi-centre clinical study and from a public healthcare system that covers all residents within the defined area. As a result, the dataset provided a complete picture of all hospitalizations and prescriptions dispensed, minimizing selection bias. There is indeed a risk of misclassification related to coding errors at the time of hospital admission, although this may not be very likely for serious conditions like bleeding and with prospective medical validation. We analyzed a population exposed to antiplatelet agents, irrespective of the indication, then including a larger scope than coronary artery disease. This is the reason why we did not consider bleeding scores devoted to coronary artery disease. Moreover, we focused on major bleeding occurring after discharge when antiplatelet agents were prescribed in secondary prevention, that is after an hospitalization for an acute thrombotic event. We did not collected early major bleeding after medical or surgical procedures. All bleeding events were medically validated using pre-specified criteria which were somewhat consistent with the "severe or life-threatening' item from GUSTO. The systematic medical validation of all bleeding limited the overestimation of incidence rates of major bleeding [23]. Exposure in our study was based on reimbursement claims data. We studied drug exposure on the basis of pharmacy dispensation but we had no information on the patients' actual intake. The lack of information on patient adherence could have led to incorrect estimations of exposure, but the clinical validation of major bleeding made it possible to check that

(A)

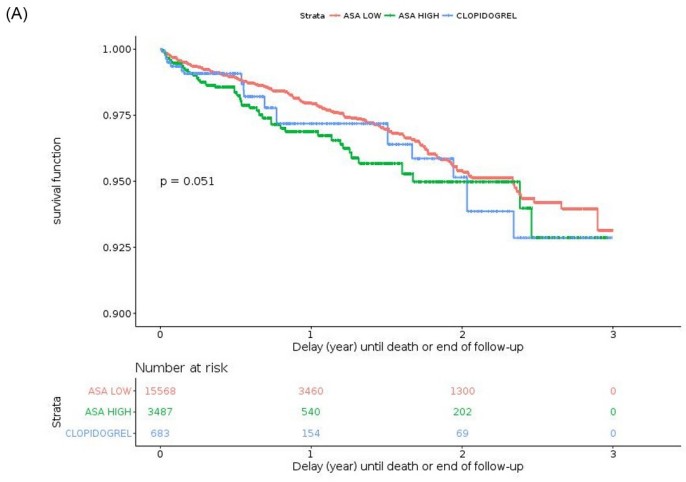

(B)

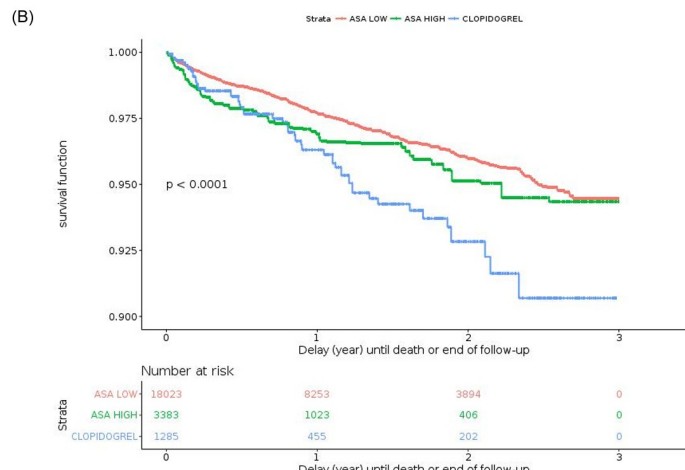

(C)

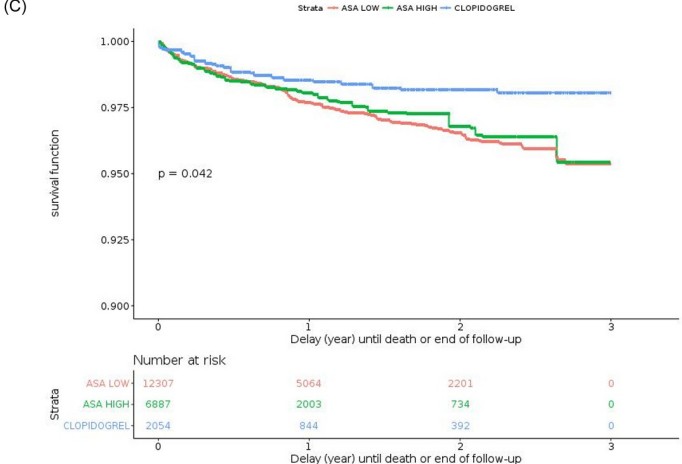

**Fig 7. Weighted Kaplan-Meyer curve for event-free survival according to antiplatelet monotherapy and indications of treatment.** Panel A: for primary prevention without cardiovascular risk factors. Panel B: for primary prevention with CV risk factors. Panel C: for secondary prevention.

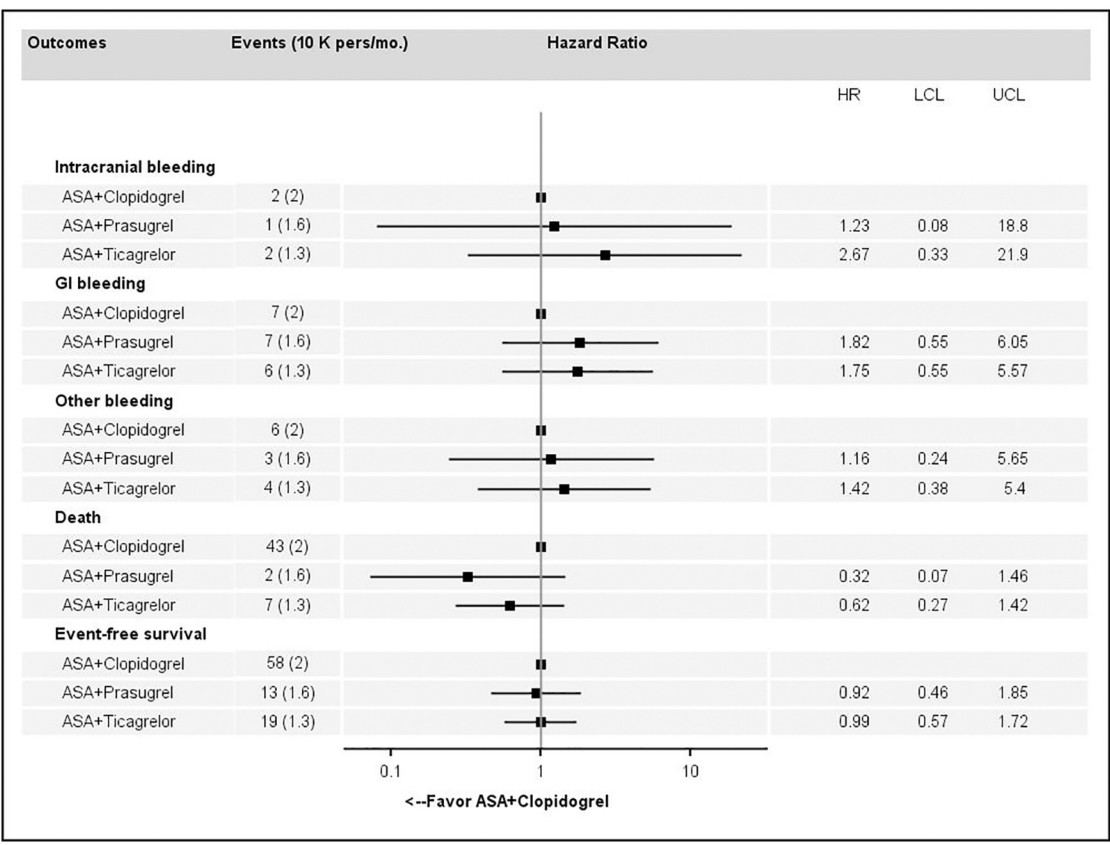

**Fig 8. Association estimated for major bleeding, death or event-free survival between antiplatelet dual regimens compared to acetylsalicylic acid (ASA) + clopidogrel.**

patients were still receiving antiplatelet drugs at the time of bleeding. Causes of death were unfortunately not available. The design and linkage to Health Insurance database allowed to state the study was exhaustive within five well-defined areas around five large French cities. But translating the result to French population is more hazardous. However, we could hypothesize that the reasons why those patients were referred to emergency department for major bleeding should not be different in other French regions. Of note, racial data cannot be collected in France. We hypothesized our population was mostly of Caucasian descent.

## Differences with other studies

Several studies have reported higher rates of major bleeding associated with antiplatelet monotherapy, especially when used for secondary prevention [9,24,25]. The meta-analysis by the Antithrombotic Trialist collaboration reported similar observations, but the authors noted heterogeneity across the studies included [13]. Several reasons could explain the discrepancy between our results and those from earlier studies.

Firstly, there was inconsistency across studies in the definition of major bleeding, whatever the type of bleeding, and in the inclusion criteria; and there were differences between randomized clinical trials (RCT) and observational studies [10]. For example, *Vazquez et al*. in a meta-analysis of 15 trials reported rates of major bleeding with aspirin ranging from 0% in 5 trials

up to 19.1% [26]. it can be noted that most of the trials included used different definitions of bleeding and 4 trials did not define major bleeding. Moreover, *Mehran et al.* [27] suggested standardized definitions for bleeding for cardiovascular clinical trials that were different from those defined by *Kaatz et al.* [19], which are closer to our criteria. GI bleeding is the main reported bleeding risk with antiplatelet drugs [28]. This observation was confirmed in the present study for all antiplatelet drugs, in monotherapy or dual therapy, but with overall lower incidence rates. This could be explained by the strict definition of the severity of major bleeding used in our study and by the systematic medical validation of all events [23].

Secondly, most studies were set in a specific clinical setting–or indication—for antiplatelet drugs, such as coronary disease [29], stroke prevention [25], peripheral artery disease [30], or primary prevention [7]. We hypothesised that bleeding risk related to antiplatelet drugs was mostly related to patient characteristics and not to their indication. Our population had an overall low comorbidity score on the adapted Charlson index, which could explain low rates of major bleeding. In primary prevention with aspirin, *Zheng et al.* in his meta-analysis reported an overall incidence of major bleeding of 23.1 events per 10,000 participant-years with a daily dose of aspirin from 50 mg to 500 mg [7]. We report higher incidence rates with high-dose ASA according to exposure time, whatever the indication.

A decrease in the bleeding risk over time has been previously reported. In a post-hoc analysis of individual data from 6 RCTs on antiplatelet therapy after transient ischemic attack or ischemic stroke, *Hilkens et al.* [31] showed high early risk of major bleeding with antiplatelet monotherapy and dual therapy, and high early risk of GI bleeding with dual therapy. No significant variations in incidence rates were found for ICH with mono or dual therapy, unlike *Li et al.* [9] who reported a reduced risk of bleeding events over time among both patients on monotherapy and patients treated initially with ASA + clopidogrel. In the present study, no similar variations were observed with clopidogrel nor with the different dual therapy regimens, probably because of a lack of power related to a small number of events.

Most studies have reported a higher risk of major bleeding with dual therapy compared to monotherapy, especially for GI bleeding [32–38]. In the meta-analysis by *Savarese et al.* [12] including 15 randomized studies, dual therapy induced a significant increase in non-fatal bleeding without an increase in ICH or fatal bleeding. These results were independent from the type of dual therapy. In our study, ICH did not significantly increase with dual therapy compared to monotherapy. We did not detect significant differences between the combinations used. These results are in line with pivotal studies on dual therapy in various indications [12,36,37,39].

Low-dose aspirin is usually defined as a daily dose from 75 mg to 325 mg. We defined two dosage regimens for ASA in monotherapy to evidence any dose-dependent bleeding risk. We thus report a lower risk of ICH and GI bleeding with ASA ≤100 mg daily compared to ASA >100 up to 325 mg daily. Similar results were reported by *Peters et al.* among patients with acute coronary syndromes, with an increased risk of major bleeding with increasing doses of aspirin [40]. For ICH, the present results have not been reported elsewhere, in particular neither by *Hilkens et al.* [31] nor by *Cea Soriano et al.* [11] Like low-dose ASA, clopidogrel increased the risk of upper or lower GI bleeding [41]. However, the CAPRIE study reported a risk of severe GI bleeding that was significantly lower among patients treated with clopidogrel (0.49%) than among patients treated with ASA 325 mg daily (0.71%) (p<0.05) [42]. This finding is in line with our observation of similar incidence rates of GI bleeding between low-dose ASA (≤100 mg) and clopidogrel.

Other major bleedings were not reported in most studies. In our study they accounted for 25% of all major bleeding events observed, mainly hematuria, oropharyngeal bleeding and epistaxis.

In addition EFS can have an interest over and above overall survival. Patients who experience major bleeding may have a lower quality of life. Surprisingly RCTs and observational studies have reported on major bleeding events and overall mortality, but not on EFS. It should be noted that the ASPREE study used a composite primary end-point including death, dementia and persistent physical disability to encompass more than bleeding events as such [43].

Whatever the antiplatelet drug used, in monotherapy or in dual therapy, we did not show any differences in mortality or EFS. The prospective registry in Dresden reported considerable in-hospital mortality after GI bleeding under antiplatelet drugs (11.9%) [44]. In most studies assessing dual therapy, no significant fatal bleedings were reported across the different regimens [12]. We evidenced lower mortality with antiplatelet monotherapy than other studies [7,8,15,16,30,42]. We do not have any rational explanation for the lower death rate in our study, especially as the characteristics of our population seem similar to those in these other studies in terms of median age and co-morbidities.

## Conclusion

Our observation in real-world clinical practice supports the best safety profile of low-dose ASA (≤100 mg daily) for secondary and primary prevention of cardiovascular diseases, and clopidogrel as an alternative for secondary prevention. We did not detect significant differences between dual antiplatelet regimens in terms of major bleeding events or death.

## Supporting information

**S1 Fig. Weighted Kaplan-Meyer survival functions for death according to antiplatelet monotherapy and indications.** Panel A: Primary prevention without risk factors. Panel B: Primary prevention with risk factors. Panel C: Secondary prevention.
(TIF)

**S1 File.**
(DOCX)

## Author Contributions

**Conceptualization:** Jacques Bouget, Emmanuel Oger.

**Formal analysis:** Frédéric Balusson, Emmanuel Oger.

**Funding acquisition:** Jacques Bouget.

**Investigation:** Jacques Bouget, Damien Viglino, Pierre-Marie Roy, Karine Lacut, Laure Pavageau.

**Methodology:** Frédéric Balusson, Emmanuel Oger.

**Supervision:** Jacques Bouget.

**Validation:** Jacques Bouget, Frédéric Balusson.

**Writing – original draft:** Emmanuel Oger.

**Writing – review & editing:** Jacques Bouget, Frédéric Balusson, Damien Viglino, Pierre-Marie Roy, Karine Lacut, Laure Pavageau, Emmanuel Oger.

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
