## [Decision Letter · Decision Letter 0]

18 May 2020

PONE-D-20-11783

Major bleeding risk and mortality associated with antiplatelet drugs in real-world clinical practice. A prospective cohort study.

PLOS ONE

Dear Dr. Rennes,

Thank you for submitting your manuscript to PLOS ONE. After careful consideration, we feel that it has merit but does not fully meet PLOS ONE’s publication criteria as it currently stands. Therefore, we invite you to submit a revised version of the manuscript that addresses the points raised during the review process.

We would appreciate receiving your revised manuscript by Jul 02 2020 11:59PM. To enhance the reproducibility of your results, we recommend that if applicable you deposit your laboratory protocols in protocols.io, where a protocol can be assigned its own identifier (DOI) such that it can be cited independently in the future. For instructions see: http://journals.plos.org/plosone/s/submission-guidelines#loc-laboratory-protocols

We look forward to receiving your revised manuscript.

Kind regards,

Timir Paul

Academic Editor

PLOS ONE

Journal Requirements:

3. Please amend the manuscript submission data (via Edit Submission) to include author Jacques BOUGET, MD, Frédéric BALUSSON, Damien VIGLINO, MD, Pierre-Marie ROY, MD, PhD, Karine LACUT,MD, PhD, Laure PAVAGEAU, MD.

Reviewers' comments:

Reviewer's Responses to Questions

**Comments to the Author**

1. Is the manuscript technically sound, and do the data support the conclusions?

Reviewer #1: Yes

Reviewer #2: Yes

2. Has the statistical analysis been performed appropriately and rigorously? 

yes

3. Have the authors made all data underlying the findings in their manuscript fully available?

Reviewer #1: Yes

Reviewer #2: Yes

4. Is the manuscript presented in an intelligible fashion and written in standard English?

Reviewer #1: Yes

Reviewer #2: Yes

5. Review Comments to the Author

Reviewer #1: Well written phase 4(claims data) study paper on an important subject in cardiovascular pharmacotherapy and care. I commend the authors for their effort in producing such a quality paper. I have few points that I would like to see if the authors can shed some light on:

- CAD is mentioned as one category but we know that there is a significant difference between stable CAD and ACS. Did we see a difference in antiplatelets complications between the two groups in the study if that was differentiated ?

- We know that Effient is more potent and causes more bleeding than other antiplatelets medications. Is there a reason why the data did not show significant differences between dual antiplatelets regimens ?

Reviewer #2: Overall, it is a good article. It is a good collection of different studies including RCTs and metanalyses.

Specific comments:

1)"Low-dose aspirin is usually defined as a daily dose from 75 mg to 325 mg" per author. However, 325mg aspirin is usually considered full dose aspirin and is NOT low dose aspirin e.g. full dose aspirin 325mg is given in Acute coronary syndromes. This part needs to be removed from the article and please add this reference regarding low dose aspirin instead: https://www.ahajournals.org/doi/full/10.1161/CIRCULATIONAHA.117.028321?url_ver=Z39.88-2003&rfr_id=ori:rid:crossref.org&rfr_dat=cr_pub%3dpubmed

PMID: 28947478

2) There are grammatical errors at multiple places. Two examples from the draft are noted below.

e.g: "It can be noted324 that bleeding was defined as reported in each study and 4 trials did not defined major bleedings." "In the present study, no similar variations were observed with clopidogrel nor

347 with the different dual therapy regimens, probably because of a lack of power related to a small

348 number of events." Need to revise grammatical errors/ rearrange sentences. Consider proper use of verbs, pronouns in the article.

3) Please define major bleeding, fatal vs nonfatal bleeding as mentioned in the article. Is there a specific/standardized score that has been used? It would be a good idea to include various bleeding scores like BARC bleeding scale, GUSTO, etc.

4) There needs a flow and and discussion seems haphazard. Please also consider adding a tabular form for articles e.g. Separately list data from different meta-analyses and from RCTs in table format as well.

5) Angers, Brest, Grenoble, Nantes and Rennes were the five cities from France from where the data was obtained. Is this data generalizable to French population vs world in general. Is there any data to support the same? Consider to use the reference below as a guide and need not be used in your article: https://www.sciencedirect.com/science/article/abs/pii/S0002914919302929

6) There have been trials that mention routine use of aspirin is not necessary for primary prevention. The data used in study is between 2012-2015. It would be a good idea to include aspirin use for primary prevention vs secondary prevention, if possible for the authors.

7) It is a known fact that low dose aspirin is low risk of bleed when compared to high dose vs dual antiplatelet therapy. Hence, not a new addition of data to the current literature. May consider to add few lines regarding anticoagulant plus antiplatelet therapy in the discussion part and can put in the limitations that it was not studied.

8) Need a flow to the article rather than presenting haphazard information. With the addition of new data, it would still be a very good article. Authors need to work on it a little more.

Thank you and best regards.

6. PLOS authors have the option to publish the peer review history of their article (what does this mean?). If published, this will include your full peer review and any attached files.

Reviewer #1: No

Reviewer #2: No

---

## [Author Response · Author response to Decision Letter 0]

29 Jun 2020

Reviewer #1: 

1. CAD is mentioned as one category but we know that there is a significant difference between stable CAD and ACS. Did we see a difference in antiplatelets complications between the two groups in the study if that was differentiated ?

Our basic idea was that the risk of bleeding is related to patient characteristics and not to the indication for treatment. Therefore, we compared incidence rates of major bleeding across exposure to antiplatelet agents, and of course, described clinical characteristics including presumed indication.

Moreover, aspirin alone was prescribed mostly in secondary prevention after a planned revascularization (stable coronary artery disease) or angina pectoris diagnosed through stress tests whilst dual regimen including aspirin was prescribed after acute coronary syndrome. Our idea was to compare monotherapies between each other on the one hand, and dual regimens between each other on the other hand.

Lastly, we used diagnosis discharge coding as well as codes for medical procedures as an attempt to identify the presumed indication for antiplatelet regimen. This was a step-by-step process more and more sensitive (and less and less specific) to minimize the number of subjects without presumed indication. 

Out of 5,674 patients classified as having coronary artery disease as presumed indication for antiplatelet monotherapy, 1,026 had been hospitalized (out of them 705 suffered from ACS) and 4,048 had ambulatory care for angina pectoris identified through stress tests. Then 705 out of 5,674 patients (12.4%) suffered from ACS.

2. We know that Effient is more potent and causes more bleeding than other antiplatelets medications. Is there a reason why the data did not show significant differences between dual antiplatelets regimens ?

We observed very few events in patients prescribed with dual regimens, keeping in mind only major bleeding were accounted for; given these low incidences we had a very low power to detect any difference.

 

Reviewer #2: 

1. "Low-dose aspirin is usually defined as a daily dose from 75 mg to 325 mg" per author. However, 325mg aspirin is usually considered full dose aspirin and is NOT low dose aspirin e.g. full dose aspirin 325mg is given in Acute coronary syndromes. This part needs to be removed from the article and please add this reference regarding low dose aspirin instead: https://www.ahajournals.org/doi/full/10.1161/CIRCULATIONAHA.117.028321?url_ver=Z39.88-2003&rfr_id=ori:rid:crossref.org&rfr_dat=cr_pub%3dpubmed PMID: 28947478

Patients were categorized as new users of either low-dose aspirin (acetylsalicylic acid, ASA ≤ 100 mg daily), or high-dose ASA (more than 100 up to 325 mg daily). We then fully agree with the reviewer and had already considered 325 mg aspirin is full dose aspirin.

2. There are grammatical errors at multiple places. Two examples from the draft are noted below. e.g: "It can be noted 324 that bleeding was defined as reported in each study and 4 trials did not defined major bleedings." "In the present study, no similar variations were observed with clopidogrel nor 347 with the different dual therapy regimens, probably because of a lack of power related to a small 348 number of events." Need to revise grammatical errors/ rearrange sentences. Consider proper use of verbs, pronouns in the article.

The revised manuscript has been reviewed by English native speaker.

3. Please define major bleeding, fatal vs. nonfatal bleeding as mentioned in the article. Is there a specific/standardized score that has been used? It would be a good idea to include various bleeding scores like BARC bleeding scale, GUSTO, etc.

Fatal bleeding according to BARC is bleeding that directly causes death with no other explainable cause. Our data only allow to detect in-hospital fatality, but we cannot state there was no other explainable cause. Defining fatal as in-hospital fatality, we observed the following number of events according to drug exposure : ASA + clopidogrel 15 events (no fatal), ASA + prasugrel 11 events (no fatal), ASA + ticagrelor 12 events (1 fatal), ASA high 53 events (7 fatal), ASA low 146 events (12 fatal), clopidogrel 13 (no fatal); overall 250 events (20 fatal).

We analyzed a population exposed to antiplatelet agents, irrespective of the indication, and then including a larger scope than coronary artery disease. This is the reason why we did not consider bleeding scores devoted to coronary artery disease.

Moreover, we focused on major bleeding occurring after discharge when antiplatelet agents were prescribed in secondary prevention, that is after an hospitalization for an acute thrombotic event. We did not collected early major bleeding after medical or surgical procedures.

All bleeding events were medically validated using pre-specified criteria which were somewhat consistent with the "severe or life-threatening' item from GUSTO.

4. Need a flow to the article rather than presenting haphazard information.

We followed a structured discussion, with sub-headings, and then we addressed some specific points: discrepancy between our results and those from earlier studies, the decrease in the bleeding risk over time, risk of major bleeding with dual therapy compared to monotherapy, difference between low- and high-dose in aspirin monotherapy, other major bleeding events (than ICH or GI bleeding), and finally mortality.

Ordering these points is a matter of debate; we thought our choice is logical; we are prepared to hear another logic provided it is explained to us.

5. Angers, Brest, Grenoble, Nantes and Rennes were the five cities from France from where the data was obtained. Is this data generalizable to French population vs. world in general. Is there any data to support the same? Consider to use the reference below as a guide and need not be used in your article: https://www.sciencedirect.com/science/article/abs/pii/S0002914919302929

Generalization to French population should be cautiously discussed. The design and linkage to Health Insurance database allowed to state the study was exhaustive within five well-defined areas around five large French cities (Angers, Brest, Grenoble, Nantes and Rennes). But translating the result to French population is more hazardous. However, we could hypothesize that the reasons why those patients were referred to emergency department for major bleeding should not be different in other French regions. Of note, racial data cannot be collected in France. We hypothesized our population was mostly of Caucasian descent.

6. There have been trials that mention routine use of aspirin is not necessary for primary prevention. The data used in study is between 2012-2015. It would be a good idea to include aspirin use for primary prevention vs. secondary prevention, if possible for the authors.

In Table 1, we showed that 40,460 patients were prescribed aspirin for primary prevention with or without cardiovascular risk factors, and 19,123 patients were prescribed aspirin for secondary prevention (mostly with coronary diseases).

7. It is a known fact that low dose aspirin is low risk of bleed when compared to high dose vs. dual antiplatelet therapy. Hence, not a new addition of data to the current literature. May consider to add few lines regarding anticoagulant plus antiplatelet therapy in the discussion part and can put in the limitations that it was not studied.

Patients exposed to anticoagulant combined with antiplatelet therapy were voluntarily not included in this analysis, because they were evaluated as part of another analysis focused on oral anticoagulant.

Bouget J, Balusson F, Maignan M, et al. Major bleeding risk associated with oral anticoagulant in real clinical practice. A multicentre 3-year period population based prospective cohort study. Br J Clin Pharmacol. 2020;1–11. https://doi.org/10.1111/bcp.14362

We add the following sentence in the "Limitation" section of the Discussion (lines 293-294):

"Patients exposed to anticoagulant combined with antiplatelet therapy were not included in this analysis."

---

## [Editor Report · Decision Letter 1]

4 Jul 2020

PONE-D-20-11783R1

Major bleeding risk and mortality associated with antiplatelet drugs in real-world clinical practice. A prospective cohort study.

PLOS ONE

Dear Dr. Bouget,

Thank you for resubmitting your manuscript to PLOS ONE. After careful consideration, we feel that the authors responded the comments but did not incorporate in the manuscript. Therefore, we invite you to submit a revised version of the manuscript as suggested below. 

A rebuttal letter that responds to each point raised by the academic editor and reviewer(s). You should upload this letter as a separate file labeled 'Response to editor'.A marked-up copy of your manuscript that highlights changes made to the original version. You should upload this as a separate file labeled 'Revised Manuscript with Track Changes'.An unmarked version of your revised paper without tracked changes. You should upload this as a separate file labeled 'Manuscript'.

We look forward to receiving your revised manuscript.

Kind regards,

Timir Paul

Academic Editor

PLOS ONE

Editor Comments :

**The authors responded to the reviewers comments but the relevant response was not incorporated in the appropriate section /discussion section except one line in limitation section.**

**Please incorporate the relevant part to the appropriate section of the manuscript.**

---

## [Author Response · Author response to Decision Letter 1]

13 Jul 2020

The relevant responses to the reviewers comments have been incorporated in the appropriate result section /discussion section.

---

## [Editor Report · Decision Letter 2]

20 Jul 2020

Major bleeding risk and mortality associated with antiplatelet drugs in real-world clinical practice. A prospective cohort study.

PONE-D-20-11783R2

Dear Dr. Oger,

We’re pleased to inform you that your manuscript has been judged scientifically suitable for publication and will be formally accepted for publication once it meets all outstanding technical requirements.

Kind regards,

Timir Paul

Academic Editor

PLOS ONE

---

## [Editor Report · Acceptance letter]

21 Jul 2020

PONE-D-20-11783R2 

Major bleeding risk and mortality associated with antiplatelet drugs in real-world clinical practice. A prospective cohort study. 

Dear Dr. OGER:

I'm pleased to inform you that your manuscript has been deemed suitable for publication in PLOS ONE. Congratulations! Your manuscript is now with our production department. 

Kind regards, 

on behalf of

Dr. Timir Paul 

Academic Editor

PLOS ONE